# Complementary and Alternative Medicine (CAM) Practices: A Narrative Review Elucidating the Impact on Healthcare Systems, Mechanisms and Paediatric Applications

**DOI:** 10.3390/healthcare12151547

**Published:** 2024-08-05

**Authors:** Patricia Anaid Romero-García, Sergio Ramirez-Perez, Jorge Javier Miguel-González, Sandra Guzmán-Silahua, Javier Adan Castañeda-Moreno, Sophia Komninou, Simón Quetzalcoatl Rodríguez-Lara

**Affiliations:** 1School of Medicine, Universidad Autónoma de Guadalajara, Zapopan 45129, Jalisco, Mexico; sjesus.ramirez@edu.uag.mx (S.R.-P.); jjmiguel@edu.uag.mx (J.J.M.-G.); sandra.silahua@edu.uag.mx (S.G.-S.); javier.adan.castaneda@edu.uag.mx (J.A.C.-M.); 2Instituto de Investigación en Reumatología y del Sistema Músculo-Esquelético (IIRSME), CUCS, Universidad de Guadalajara, Guadalajara 44100, Jalisco, Mexico; 3Departamento de Investigación, Instituto Cardiovascular de Mínima Invasión (ICMI), Zapopan 45116, Jalisco, Mexico; 4Unidad de Investigación Epidemiológica y en Servicios de Salud, CMNO OOAD Jalisco Instituto Mexicano del Seguro Social, Guadalajara 44160, Jalisco, Mexico; 5Faculty of Health and Life Science, Swansea University, Swansea SA2 8PP, UK; sophia.komninou@swansea.ac.uk

**Keywords:** complementary and alternative medicine, paediatric health, traditional medicine, well-being, physiological, physical, mental health, nutritional, psychological

## Abstract

While research on complementary and alternative medicine (CAM) for the general population is expanding, there remains a scarcity of studies investigating the efficacy and utilisation of CAM practices, specifically in the paediatric population. In accordance with the World Health Organization (WHO), the prevalence of the parental utilisation of CAM in their dependents is estimated to reach up to 80%. This literature review identified broad, heterogeneous, and inconclusive evidence regarding CAM’s applications and effectiveness, primarily attributed to variance in sociodemographic factors and differences in national healthcare systems. Additionally, the review identified a lack of consensus and polarised positions among mainstream professionals regarding the mechanisms of action, applications, and effectiveness of CAM. This narrative review presents varied results concerning the efficacy of most CAM therapies and their applications; however, some evidence suggests potential benefits for acupuncture, yoga, tai chi, and massage in improving physical and mental health. Moreover, the available evidence indicates that meditation may enhance mental health, while reiki may only influence patients’ perceptions of comfort. In light of the intricate and multifaceted nature of herbal medicine, it is imperative to assess its efficacy on a case-by-case basis, taking into account the specific compounds and procedures involved. This comprehensive review serves as a valuable resource for health professionals, offering guidance for personalised healthcare approaches that consider the values and beliefs of patients, thereby facilitating integrated, evidence-based practices aimed at enhancing the quality of healthcare services and patient satisfaction.

## 1. Introduction

Healthcare systems can be classified into two main categories: (1) conventional medicine (also referred to as modern, western, mainstream, or allopathic) and (2) traditional medicine (encompassing indigenous medicine and complementary and alternative medicine) [1]. Conventional medicine can be understood as a system that relies on a western medical concept that prevents, treats, and rehabilitates symptoms and diseases through evidence-based practices, including drugs, radiation, or surgical procedures [2]. Complementary and alternative medicine (CAM) has been defined by the World Health Organization (WHO) [3] as a broad set of healthcare practices that are not part of the conventional health system practices and are not fully integrated into the dominant healthcare system [3].

However, controversies exist regarding the terminology given the lack of homogeneous international health policies and the differences amongst national healthcare systems and cultural perceptions, particularly applicable to non-western countries. For instance, the national health system in China has two variables equally accepted and supported by governmental policies and regulations: western medicine and traditional Chinese medicine [4,5,6,7]. As traditional Chinese medicine is a key element of the national health system, practices encompassed within traditional Chinese medicine (such as acupuncture, tai chi, and herbs) are not perceived as CAM practices but as mainstream practices [6,7]. Thus, for the purposes of this review, the main difference between conventional and unconventional medicine is whether the practice is formally integrated into the dominant healthcare system of a nation (conventional) or not (unconventional).

These definitions have shown a dynamic pattern, fluctuating throughout the years; between 1990 and 2013, the terms “complementary” and “alternative” were prevalent in the literature as separate terms [8]. In addition to the aforementioned terms, the concept of “integrative medicine” has a notable role within this topic. It is considered integrative medicine when both conventional and unconventional medicine are combined [8]. The term CAM, which includes both complementary and alternative medicine, has become more acceptance since 2000, and it is now the most widely recognised term [8]. As separate terms, “complementary” refers to methods and practices used together with conventional medicine, whereas “alternative” describes practices and methods used as substitutes to conventional medicine [9]. Acknowledging these terminological differences, CAM consists of a set of medical practices, products, or systems, but the approaches to these unconventional practices differ and are still largely unregulated globally [10]. Based on these concepts, a CAM practitioner can be defined as someone who delivers a range of healthcare services and therapies that are beyond the scope of conventional medical treatment; in certain western countries, such as the UK, according to the National Healthcare System (NHS), no statutory regulations exist applicable to CAM practices (except for chiropractors). Nevertheless, some CAM practitioners, especially in western countries, are accredited by specific professional standard authorities pertaining to health and social care services or voluntarily chosen professional associations. These associations evaluate qualifications and provide registers to practice CAM to their specific standards. However, it should be noted that holding a medical degree is not mandatory to practice CAM, yet the percentages of CAM practitioners holding both qualifications vary from 8 to 25% depending on sociodemographic factors [11,12,13].

CAM practices have been present in healthcare systems worldwide since ancient times [14]. It has become a global tendency to use CAM, with prevalence rates ranging from 8 to 76% in the general population [15]. Regarding the paediatric population, in 2013, an estimated 52% of European children used CAM therapies [16], and the available data indicate an increase in the popularity of CAM methods worldwide [17]. Likewise, the WHO estimates that parents’ use of CAM therapies for their children varies geographically, ranging from 10% to as high as 80% [18].

Several studies have published favourable results regarding the use of CAM; however, non-rigorous and unclear methodologies have been identified. Such inconsistent results contrast other scientific evidence following rigorous methodologies [19]. Moreover, differences in CAM’s prevalence, purpose, and satisfaction measurements pose a significant challenge due to the variability of the findings across distinct study designs and countries [20,21,22].

The heterogeneity in existing trials, together with inconclusive results about CAM’s effectivity and safety, has led health providers and scientists to adopt different positions regarding whether to recommend CAM or not, creating, at the same time, confusion amongst CAM users in the general population [19]. This heterogeneity within healthcare providers hinders the formality in reporting adverse events. Despite the many-sidedness, the number of case reports of adverse events is growing and varies from mild and self-limited effects to fatal outcomes leading to death [23]. Hence, concerns regarding patients’ well-being have arisen within the scientific community, and CAM interventions’ effectiveness and safety have been highly questioned. This lack of consensus regarding the use of CAM affects primarily vulnerable populations, such as the elderly and children, given that their health is mostly dependent on their caregivers and guardians, who rely on different sociocultural and geographical contexts. Thus, high-quality research examining the effectiveness and safety of CAM therapies is crucially needed.

This review aims to provide an overview of CAM around the globe, as well as to explore the applications, mechanisms of action, and adverse effects of CAM therapies in the paediatric population.

## 2. Methodology

This comprehensive narrative literature review employed Google Scholar, PubMed, and the Cochrane Library for academic searches, using keywords and topical research. Initially, broad terms like “complementary and alternative medicine in children” were used, but, later, they were refined to specific topics. Eligible articles included systematic reviews, meta-analyses, randomised controlled trials (RCT), observational studies, literature reviews, case reports, and discussion articles. Only English language studies published between 2010 and 2024 were considered, with exceptions made for relevant older studies or when recent data were lacking. Preclinical studies were included to explore mechanisms of action, and there were no geographical limitations, so as to ensure a global perspective on CAM and children. When a lack of paediatric evidence was identified, considerations were given to studies performed in animal and adult populations, as there are often common underlying mechanisms and therapeutic strategies (Table 1).

## 3. National Healthcare Systems and CAM

The architecture of a typical healthcare system is contingent upon distinct factors, each intrinsic to the particular country in question. These factors encompass the climate and natural environment, geography, population density and demographics, history, sociocultural ideologies, and political and economic status [24]. These elements collectively delineate a unique framework and national context that significantly shapes healthcare systems’ governance and financial structuring. National authorities subsequently formulate criteria and strategic frameworks for resource allocation, with the primary goal of facilitating the provision of healthcare services, including health promotion initiatives, disease prevention measures, curative interventions, rehabilitation programs, and palliative care [24]. The dissemination of these services to the population at large is facilitated through the deployment of healthcare personnel, the availability of medical products, adherence to established service standards, and the dissemination of pertinent health-related information [24].

CAM occupies a significant role within the sociocultural ideologies that mould and influence national healthcare systems across various levels. Epidemiological data reveal a notable global surge in CAM utilisation [25,26]. Notably, in Europe, research indicates that approximately 29% of the population used CAM therapies in 2014 [27]. Similarly, in the United States of America (USA), CAM usage escalated by 17.5% between 2002 (19.2%) and 2022 (36.7%) [28].

This trend may have influenced government resource allocation, prompting augmented funding for CAM research; illustrating this financial impact, in the USA, the National Institutes of Health (NIH) report highlights a substantial increase in budgetary provisions for the National Center for Complementary and Integrative Health (NCCIH), whose NIH congressional appropriation has increased by approximately 40% since the last decade (121,373 dollars in thousands in 2013 vs. 170,384 in 2023) [29]. However, in 2019, the NCCIH’s total reported budget was 517,246 dollars in thousands, which represents an increase of 253% from the congressional appropriation reported in the same year (146,473 dollars in thousands) [30].

In some other countries, the increased prevalence of CAM may not only have impacts on the financial resources of a healthcare system, but it also has elicited changes in particular settings, such as cancer. In Malaysia, where the prevalence of CAM users was approximately 46.5%, a multi-centre cross-sectional study reported that, in patients with breast cancer, the use of CAM was associated with delays in the presentation (odds ratio [OR] = 1.65; 95%, confidence interval [CI]: 1.05–2.59), diagnosis (OR = 2.42; 95% CI: 1.56–3.77), and treatment of breast cancer (OR = 1.74; 95% CI: 1.11–2.72) [31]. Similar results have been reported in the paediatric population; according to an Indonesian study on a cohort of oncologic paediatric patients, the authors reported an association between CAM users and a delay in the diagnosis of solid tumours (OR = 1.86, 95% CI: 1.13–3.08, *p* = 0.015), with a median delay of 14 days [32].

Regardless of the objectivity of these data, it is worth considering western medicine’s heterogeneous national health coverage, particularly in remote areas where CAM is the most available or affordable healthcare. The integration of well-established policies and strategies for CAM regulation in the crucial infrastructure components of national healthcare systems may contribute to advancing the particular health system in terms of quality, safety, and efficacy.

Despite an increase in CAM usage in the paediatric population [33,34], little has been investigated concerning its global impact, which indicates the need for the further exploration of this healthcare concern.

## 4. Paediatric Population and CAM

The prevalence of CAM therapies in paediatric populations varies widely according to the cultural and geographical context; for instance, in South Korea, where Korean medicine—a form of East Asian traditional medicine—plays a central role within South Korea’s healthcare system, 8.13% of the population under 19 years old uses integrative medicine [35]. Still, it is estimated that, around the world, the overall rates of CAM use in children range from 10.9 to 87.6% depending on the country and the sociocultural factors of the parents [36].

Lately, it has become widespread practice for parents to use CAM therapies for their children rather than conventional medicine [37]. In previous years, in the USA, the number of children who underwent consultations with CAM providers was higher than the number of children taken to conventional practitioners [38]. In the past, the estimated amount spent by American parents on CAM therapies for their children annually in the whole country was more significant than the amount spent on hospitalisation (USD 13.7 billion, compared to USD 12.8 billion, respectively) [38].

Currently, studies have shown that vulnerable children, such as those with chronic, recurrent, or incurable conditions that are difficult to manage or children taking prescription medication [39], are more likely to seek CAM alternatives than others [40,41,42]. The rates of children with such conditions are estimated to be greater than 50% of the total number of children using CAM, with oncology patients leading the list with 63% [43,44]. Other common conditions that are found to be linked to the higher use of CAM are asthma and mental health conditions such as autism spectrum disorder, attention deficit hyperactivity disorder (ADHD), depression, and anxiety [40,45]. Furthermore, parents predominately choose CAM to treat specific symptoms among their children, such as headaches, neck or back pain, colic, and respiratory symptoms like chest colds [17,45], runny nose, cough, and sore throat [46].

Despite the large number of studies supporting the strong link between CAM usage and chronic illnesses, some studies have not found a correlation in the therapeutic approach. Instead, they suggest that CAM is used for children to increase their physical fitness [47], prevent illness or maintain good health, and deal with symptoms such as vomiting or abdominal pain [39]. Unlike four- to eleven-year-olds, for whom the main reason for the use of CAM is to treat vomiting or abdominal pain, adolescents aged 11 to 17 are more likely to use CAM to control their weight and improve their personal image or body shape [17,47,48].

## 5. Factors Influencing CAM Usage in the Paediatric Population

The factors that lead caregivers or legal guardians to use CAM can be classified into the following categories: sociocultural, economic, and educational [36,49,50].

### 5.1. Sociocultural Factors

The use of complementary therapies strongly relies on the sociocultural and ethnic context. In European countries, where the official national healthcare system is mainly based on a western medical approach, the CAM use rates among children range from 25 to 50% [51]; in contrast, traditional Chinese medicine (TCM) is a pillar of the Chinese national healthcare system, and the prevalence of parents using CAM for their children is by far the highest in the world, reaching approximately 76%, with rates as high as 96% in paediatric oncologic patients [52,53].

Furthermore, some caregivers decide to use CAM in their dependents simply because it is an inherited tradition [54], whereas some other caregivers rely on the conviction that CAM methods or therapies are natural or use natural components; hence, they are innocuous and safe [54]. It has also been reported that children whose parents/caregivers are CAM users are more likely to use CAM in their descendants/dependents as well [39,55,56]. Some parents choose CAM therapies to be considered “good parents” [54], which eases their sense of agency over their offspring. For other parents, choosing CAM for their children is related to superstition and beliefs [57].

Parents/caregivers commonly decide to use CAM in cases where their children face chronic, incurable, or terminal pathologies. Therefore, guardians feel helpless, and the active and personal seeking of an alternative to improve their children’s condition becomes imperative [58]. Moreover, it has been observed that parents decide to use CAM when they lack an accurate diagnosis of their child’s condition. Consequently, parents’ decision to give their child CAM is often taken out of despair to find a solution to their child’s problem, based on a feeling of failure and concerns related to the side effects of conventional medical treatments [54].

As previously explained, cultural factors highly influence the selection and transmission of CAM therapies; this inherited “CAM culture” promotes the seeking and use of these therapies in each generation, consequently impacting the national health system.

### 5.2. Economical and Educational Factors

It is generally accepted that broad access to CAM and its affordability for the general population account for some of the primary causes of CAM use [15]. Nevertheless, controversies exist regarding the association of CAM, education, and income. On the one hand, some academics support the idea that easy access to CAM and the low costs of such interventions are associated with higher rates of CAM use [54,57]. On the other hand, the existing literature suggests that people with a higher educational level, who tend to have higher incomes, are more likely to afford to become CAM users [35,57]. Still, some evidence shows that a higher educational level outweighs wealth in the decision to use CAM; however, the available literature supporting any of the aforementioned perspectives is insufficient to favour one over another.

Despite the lack of consensus regarding the role of educational and income levels in influencing the incidence and prevalence of CAM use, there is a growing body of objective evidence suggesting that these two factors are strong determinants in selecting the type or modality of CAM. For example, acupuncture and relaxation were associated with higher educational levels, whereas chiropractic care was associated with lower educational levels [59]. Thus, chronic illnesses and children’s poor health prognoses, along with the household income, educational level, and socio-geographical factors, altogether account for the major factors that encourage parents/caregivers to use CAM therapies for themselves and their dependents.

In an attempt to construct a western prototype of a typical CAM child patient, some literature reflecting data from the USA and Germany states that the classical paediatric patient under CAM interventions would be a non-Hispanic white adolescent living in a house with high income and whose parents have a higher educational level [40,42]. These findings were also confirmed by the National Health Statistics Report in the USA (2015), where the paediatric populations with the lowest prevalence of CAM usage were Hispanic and African American children [17].

Whatever the reasons or motivations that lead parents or adolescents to seek CAM, the high rates of its popularity urge us to examine whether these interventions are indeed effective and safe.

## 6. CAM Classification

The therapies included in the definition of CAM are deliberately broad since practices considered complementary or traditional in one country or a particular culture might be regarded as part of conventional medicine in a different one [59]. Thus, several classifications of the therapies considered as CAM have been proposed by different scholars. The NCCIH proposes three categories based on their therapeutic input: physical (e.g., massage, acupuncture), nutritional (e.g., probiotics, herbal therapy), and psychological (e.g., hypnosis, spiritual healing/prayer) [60].

However, Mollaoğlu and Aciyurt [61] suggested the following classification: (1) alternative and medical system (AMS)—acupuncture and homoeopathy; (2) mind–body interventions (MBI)—relaxation techniques, imagery, spiritual healing/prayer, biofeedback, and hypnosis; (3) biologically based therapies (BBT)—herbal therapy and dietary supplements; (4) manipulative and body-based methods (MBBM)—massage therapy, exercise, and chiropractic or osteopathic care; (5) energy therapies (ET)—energy healing and reiki.

Considering that the vast majority of CAM interventions share input features, one single CAM intervention may fall within different categories; hence, the following adjusted classification and categorisation are proposed (Figure 1).

Then, it is vital to determine the specific features and mechanisms of action of each type of CAM, given that each intervention may distinctly impact the physiological and psychological domains of each CAM user.

## 7. CAM Mechanisms of Action, Applications, and Adverse Effects

### 7.1. Acupuncture: Overview and Mechanism of Action

Acupuncture is an ancient Chinese medicine component that works with the “chi” (qi) or vital energy flowing in channels/meridians through the human body [62]. According to this practice, a body becomes sick when the flow of “chi” is blocked or disrupted. Hence, acupuncture aims to restore the vital energy flow, stimulating certain meridian points [43,44].

Acupuncture is widely used around the world, producing numerous variations of the acupuncture therapy technique and its principles. However, the most used method is the collocation of solid-core and small needles that penetrate the skin, collocated alongside the body and stimulated manually [40]. In TCM, the stimulation of the needling sites, or acupoints, leads to the sensation of numbness, distension, or tingling at the needling site [63]. This sensation is referred to as “de-qi”, which radiates along to the corresponding meridian, inducing the desired effect. Acupressure is a different modality that interacts with the vital energy flowing in the meridians using the hands or fingers of the therapist. It has its origins in China and is based on the same principle as acupuncture [59].

The effects of acupuncture are explained mainly by the diffuse noxious inhibitory control (DNIC) that involves the activation of the first neuron through the mechanical, thermic, or chemical stimulation of nociceptors and polymodal receptors (PMR) [64]. Primary nociceptive neuron depolarisation propagates through A-δ and C-afferent fibres, whereas PMR signals act at the dorsal horn through A-β, A-δ, and C-afferent fibres [65,66,67]. These first neurons will then activate secondary afferent neurons in the spinal cord, depolarising tertiary neurons in superior brain structures such as the thalamus and the hypothalamus [63,65].

The local application of the needle primarily activates nociceptors and PMR, but it can also activate local vascular processes and inflammatory responses (e.g., mast cells) [64,68]. The literature demonstrates that several molecules are involved in the positive effects of acupuncture [65,68,69]. For instance, P2 × 3 and TRPV1 receptors play a significant role in nerve conduction. In contrast, the inflammatory response is mediated by mast cells, microglia, astrocytes, and oligodendrocytes expressing OX-42, matrix metalloproteinase (MMP)-9/MMP-2, TNF-α, IL-1β, IL-6, and IL-10. Lastly, cell adaptation relies on the activation of CXCL1, p38MAPK, ERK, and C-Jun/N-terminal kinase signalling [63,64,70].

Regarding the modulation of pain, mood, and behaviour in the CNS, acupuncture induces positive effects, most likely by the release of endogenous opioids (e.g., endorphins, enkephalins, dynorphins, nociceptin/orphanin) and neurotransmitters (e.g., 5-hydroxytryptamine, noradrenaline, dopamine) [63,71,72,73].

#### Applications and Adverse Effects

Acupuncture is used to treat and manage a vast number of conditions and illnesses. Acute pain (e.g., postoperative pain, labour pain, dysmenorrhea, and migraine), chronic pain (e.g., low back pain, osteoarthritis, knee pain, neck pain, chronic shoulder pain), and behavioural conditions such as insomnia, anxiety, and depression are amongst the most common indications in the general population for the use of acupuncture [69].

Unlike adults, because children often fear needles, acupuncturists stimulate acupoints and meridians using non-needle methods, such as heat, magnets, lasers, and massage [43,44]. Thus, acupressure is commonly used for children [40,74]. Among the long list of conditions and illnesses in children treated with acupuncture, some studies have suggested positive outcomes for acupuncture. A systematic review evaluating RCTs reported that acupuncture significantly reduced the incidence of nausea and vomiting in patients undergoing surgery [75]. Similarly, some studies, such as a double-blind RCT conducted in children with headaches, have found laser acupuncture to be superior to placebos in decreasing the number of headaches per month and their severity [76]. Additionally, clinical guidelines such as NICE in young people and adults in the UK recommend acupuncture as a prophylactic treatment for primary headache disorders such as tension-type headaches and migraine [77]. Positive outcomes for acupuncture and conditions such as infant colic, nocturnal enuresis, dysmenorrhea, pain management, and allergic rhinitis have been described in other reviews [78].

Regarding the safety of acupuncture for children, systematic reviews have shown that it is usually safe when practised by licensed and trained acupuncturists who follow studied and detailed protocols [79,80,81]. Furthermore, the severe adverse effects of such practices are limited, suggesting that it is a relatively safe practice [40,82,83]. Confirming that acupuncture is a safe practice, Adams et al. (2011) [82] found, in a systematic review, that most of the adverse effects were mild and included crying, pain, bruising, numbness, and haemorrhage in the site where the needles were placed. Moreover, dizziness and nausea were reported, whereas severe adverse effects were limited and linked to technical errors and unskilled practitioners rather than risks related to the procedure itself [82]. Other severe adverse effects were reported, including pneumothorax, nerve injury, vasovagal reactions, and, in some cases, cross-infection when the needle was reused with an improper sterilisation technique [62,64].

### 7.2. Massage and Bodywork: Overview and Mechanism of Action

Massage therapy is an ancient Chinese practice defined as the manipulation of the soft tissue and muscles of the human body. It aims to enhance the patient’s emotional and physical health [83]. Massages are most commonly performed on the acupuncture/meridian points [83]. However, the alternatives encompassed in massage and bodywork vary widely depending on the cultural and geographical background, since most cultures have their own traditional massage methods [43,44].

Massage is believed to heal by raising the temperature in the manipulated tissue by promoting the targeted tissue’s metabolism. This causes the dilation of the capillaries and speeds up the circulation of the blood and lymph. It also enhances the supply of nutrients in the surrounding muscles and tissue, promoting, at the same time, their growth and development [84,85].

The general physiological effects of massage remain unclear, but a possible explanation is based on the increase in blood circulation, a decrease in cortisol levels, and an increase in the endogenous levels of serotonin; hence, a relaxation stage is reached, loosening the muscles and joints and enhancing the overall feeling of well-being [43,44].

#### Applications and Adverse Effects

Massage therapy is offered as support for a broad spectrum of health conditions; it has been reported that massage reduces gastrointestinal issues such as constipation and acute diarrhoea [86], ameliorates asthma symptoms, reduces pain, and improves vascular circulation [43,44].

The available evidence, including systematic reviews and meta-analyses, strongly supports massage care as a suitable treatment for infants to treat conditions mainly related to improving growth rates and weight gain in premature infants [87,88,89,90,91]. For instance, an RCT conducted on Indian neonates found that massage therapy could help premature infants gain around 476.76 g within 28 days in newborns with a birth weight of less than 1800 g [92]. The stimulation of the vagal activity explains the beneficial effects of massage on newborns’ weight gain. Hence, the levels of insulin-like growth factor-1 (IGF-1) increase, as well as their gastric motility; consequently, the maturation of the sleep–wake state behaviour and an increase in the infant’s weight occur [91].

Additionally, promising results were found in a meta-analysis that analysed 26 studies encompassing 2644 paediatric patients with a diagnosis of acute diarrhoea, where it was shown that massage care was clinically more effective than pharmacotherapy in treating paediatric acute diarrhoea (*p* < 0.01) [83]. The exact physiological mechanism by which massage might clinically improve acute diarrhoea is still not fully elucidated. Still, it is hypothesised that massage, as described before, enhances gastric motility and influences the secretion of hormones, cytokines, and gastric acid. Consequently, the spontaneous movements of the bowels are rhythmically enhanced [83].

Moreover, in favour of massage, evidence suggests that it is a promising treatment for jaundice in newborns. An RCT studying 40 newborns at 34 to 36 weeks of gestation, without any disease at birth, found a decrease in the transcutaneous bilirubin levels of premature neonates who received massage interventions starting from the third and fourth day (*p* = 0.003 and *p* < 0.001, respectively) when compared with the control group. Similarly, statistically significant differences were noted in terms of increasing stool frequency in favour of the massage group (*p* = 0.002) [93]. The decreased levels of bilirubin and the increase in stool frequency were explained again by the stimulation of the vagal nerves and the stimulation of gastric motility, which is caused by the release of gastrointestinal hormones, specifically gastrin and cholecystokinin, due to the stimulation of the peripheral nerves. Consequently, digestion is accelerated and, therefore, the excretion of bilirubin through the stool [93].

Concerning pain management, an RCT performed with 52 hospitalised children aged ten to 18 years suggested that massage therapy is useful in reducing the overall pain of children with cancer after each massage session (*p* < 0.001), as well as decreasing the pain related to walking (*p* < 0.05) [85]. The pain improvement was measured using the Visual Analogue Scale (VAS). The positive effects of massage on pain have been attributed to the enhancement of blood circulation and lymphatic flow, as well as reduced levels of stress-related hormones to induce a state of body relaxation and increase muscle tone and range [85]. However, this was a single-blinded study limited to a small sample. Hence, more research exploring children’s pain management is needed before drawing conclusions.

In summary, massage’s most robust evidence suggests that this practice increases weight gain in premature infants [87,88,89,90,91] and also reduces anxiety and suffering in hospitalised children and adolescents with psychiatric disorders [43,44].

### 7.3. Mind–Body Therapies

Mind–body therapies (MBT) can be generically conceptualised as any therapy that combines the brain, mind, body, and behaviour with the purpose of healing [94]. These practices acknowledge that emotional, mental, social, and spiritual factors directly impact health. MBT includes tai chi, qigong, yoga, meditation, and other types of relaxation (e.g., biofeedback, neurofeedback) [95,96].

Given that MBT encompasses a broad variety of practices, describing a unique mechanism of action is challenging; thus, the molecular and cellular effects of these practices on health will be described separately. Based on the available information, yoga, meditation, and tai chi will be covered.

#### 7.3.1. Yoga: Overview and Mechanism of Action

Yoga is an ancient Hindu practice that has become more popular in western countries in recent years. Yoga is generally used as a form of exercise and movement, whereas therapeutic yoga works by uniting the mind, the body, and the spirit through conscious breathing and body postures [97]. It is believed that yoga influences the parasympathetic nervous system, reducing cortisol levels [98] and the heart rate [97].

In 2012, according to the National Health Survey, in the USA, yoga was positioned among the top five most commonly used CAM methods for children between two and 17 years old; its prevalence increased from 3.1% in 2012 to 8.4% in 2017 [17]. Within the paediatric population with mental health conditions, yoga was found to occupy the third place among the most common CAM therapies [52].

Children facing mental health difficulties may seek yoga because of the essential effects that it has on psychological functioning. From a physiological perspective, RCTs have suggested that the positive impact of yoga might be linked to the effects on the sympathetic nervous system and the regulation of the hypothalamic–pituitary–adrenal axis. Systemically, yoga decreases the serum levels of IL-6, TNF-α, and glucocorticoids [94].

The aforementioned systemic changes can be molecularly explained by a yoga-induced decrease in NF-κB activity and the activation of glucocorticoid receptors; as a consequence, the heart rate decreases, tissular perfusion increases, and the overall mood is enhanced [40,94].

##### Yoga: Applications and Adverse Effects

Having explained the physiological mechanism of yoga, it is not surprising that a body of evidence evaluating systematic reviews, meta-analyses, and RCT interventions of yoga in children has indicated positive effects mainly in behavioural conditions, the control of anger, anxiety, stress, body dissatisfaction, control over emotions, self-esteem, and resilience [40,98].

Apart from enhancing mental well-being, a systematic review analysing 15 trials identified yoga as a beneficial treatment for specific paediatric conditions, such as asthma, irritable bowel syndrome (IBS), eating disorders, juvenile idiopathic arthritis, and fibromyalgia. Furthermore, it was found to improve some metabolic and hormonal parameters and chronic pain [99].

The overall benefits of yoga for conditions like IBS and eating disorders have also been mainly related to the decrease in anxiety and control over negative emotions caused by the course of the condition. For instance, an RCT evaluating the effect of yoga on 54 patients aged 11 to 21 years old with diagnosed eating disorders found decreased scores in their anxiety, depression, and food preoccupation. Consequently, the enhanced mental well-being facilitated total clinical recovery from preexisting eating disorders [100].

Some studies conclude that yoga might be promising; for instance, an exploratory and experimental study performed on children with autism spectrum disorder (ASD) reported that yoga interventions significantly reduced sleep, gastrointestinal, and behavioural problems [101]. Regarding paediatric cancer, pilot non-RCTs have supported the safety and efficacy of yoga [102]. At the same time, some other systematic reviews show inconclusive results in recommending yoga as an effective mental health intervention in children, given the identified heterogeneity in the measured outcomes and the diverse and non-comparable populations [103,104].

When assessing yoga’s safety, the existing literature identifies a few adverse effects of this practice. According to evidence, yoga and MBT are considered relatively safe practices in the general population [40]. A systematic review and meta-analysis of RCTs investigating the safety of yoga found that 57% of the trials did not report any adverse effects, and, from the reported adverse events, only 2.2% were classified as intervention-related. Among these effects, 10.9% were non-serious events (e.g., musculoskeletal injuries), and 0.6% were severe events (e.g., permanent physical disabilities and injuries needing surgical interventions). The systematic review also reported a low bias risk and low heterogeneity among the trials [105]. Similarly, more systematic reviews have concluded that yoga is safe and effective [105,106].

Considering the evidence mentioned above, it can be concluded that yoga is a safe and cost-effective practice that can be a reasonable method to improve overall well-being and enhance mental health conditions, stress, and anxiety in children.

#### 7.3.2. Meditation: Overview and Mechanism of Action

Meditation is one of the mind–body therapies that originally came from traditional contemplative practices used in ancient cultures under a spiritual basis with a focus on gaining body consciousness [97].

Meditation encompasses different practices (Table 2) that generally focus on concentration and attention. However, providing a specific definition is challenging since there is a wide variety of techniques and types, each with its own characteristics and definitions. However, all forms have a common component: training the mind [107].

As observed with the previously discussed CAM interventions, meditation techniques have recently gained popularity. In the USA, for example, in 2012, approximately 0.6% of children aged 4 to 17 years practiced any form of meditation, whereas, in 2017, 5.4% of children were found to use meditation [109].

Unlike ancient meditation practices, nowadays, meditation has been incorporated into modern practices and cultures beyond spirituality and religious settings [107]. Within modern meditation practices, and despite the different techniques used to meditate, they all fundamentally increase the awareness and the sense of self through intentional practices such as breathing, listening, specific movements, and sounds [108].

Little is known about the physiological effects of mediation practices among children. Still, a systematic review and meta-analysis suggests that meditation decreases physiological stress-related substances, such as cortisol, C-reactive protein, triglycerides, and TNF-α. The relaxation state induced by meditation decreases the heart rate, blood pressure, and expression of inflammatory cytokines, leading to feelings of well-being [110].

##### Meditation: Applications

The positive effects of meditation on children’s mental well-being have drawn attention to the possible impact on children’s cognitive functioning and behavioural improvements, specifically regarding mindfulness [111]. A single-blinded RCT study conducted on 32 Italian children, comparing mindfulness-oriented meditation (MOM) versus active control conditions without meditation, found that mindfulness could effectively improve cognitive, emotional, and social abilities in children [111]. The importance of such results centres on the possibilities of reaching clinical improvements in mental conditions such as ADHD, autism spectrum disorder, and behavioural disorders among children, which are generally difficult to treat with mainstream medicine [111]. Focusing on children with ADHD, a systematic review and meta-analysis evaluating RCTs reported that mindfulness-based therapies improved ADHD symptoms (*p* = 0.006). However, significant heterogeneity in the measured outcomes (e.g., effects on internalising behavioural problems, externalising behavioural problems, parental stress) was also reported, highlighting the overall complexity when interpreting the possible benefits of mindfulness-based interventions [112].

Moreover, some authors have suggested that meditation should be included in academic curricula to help children to improve their attention and help those with behavioural difficulties [107,108]. However, research exploring the link with improved cognitive function has only recently emerged. Hence, more trials are still needed to analyse the benefits of mindfulness. Despite being promising, the available evidence still does not confirm the effectiveness of mindfulness for such conditions [111,112].

#### 7.3.3. Tai Chi: Overview and Mechanism of Action

Tai chi is a mind–body exercise rooted in Chinese culture; it directly translates to “grand ultimate” and is considered a philosophical and theoretical practice that seeks balance (qi balance) between interactive dualistic phenomena through meditative movement. The practice involves dynamic dance-like postures, relaxation, and breathing control [113,114].

Traditional tai chi has three fundamental principles: body, breath, and mind focus. Typically, tai chi consists of a complex series of sequenced and choreographed movements, meditation, visualisation, and deep abdominal breathing techniques that induce a relaxation state, leading to health enhancement [113].

It has been suggested that tai chi has multiple central and peripheral effects, mainly influencing the autonomous nervous and immune systems [115]. However, tai chi’s effects are not limited to the systems mentioned earlier; it has been reported that tai chi practices also impact neurological function [116] and the musculoskeletal system in children and the young population [80,117,118].

As a consequence of the stimulation of the autonomic nervous system, tai chi reduces the heart frequency and improves tissue perfusion; the latter effect is mediated by both the recruitment of reserve capillaries and an increase in the VO_2_ peak [119]. The neurological effects are mediated by the modulation of neurotransmitters (e.g., dopamine, serotonin, and adrenaline) and neuromodulators (e.g., endorphins, brain-derived neurotrophic factor (BDNF), and insulin-like growth factor (IGF)), which, in conjunction, can elicit neuroplasticity with structural and functional brain changes [80].

Regarding the immune system, tai chi has been shown to decrease inflammatory cytokines (e.g., IL-6, IL-8) and other inflammatory markers (e.g., cortisol and C-reactive protein) [80,115]. Additionally, it has been reported that, after tai chi is practised, monocytes decrease the expression of TNF-α and IL-6; the described modulation is also associated with the upregulation of the interferon regulatory factor (IRF) family and the downregulation of NF-κB transcription factors, resulting in the modulation of the inflammatory response [94].

Moreover, tai chi also exerts effects on oxidative stress, including an increase in glutathione peroxidase, catalase, and superoxide dismutase levels. Such enzymes, classified as scavengers, positively enhance the total antioxidant capacity [120].

##### Tai Chi: Applications

In the general population, tai chi has been used to treat a wide variety of conditions and illnesses. The existing scientific evidence suggests that tai chi is useful in decreasing pain, increasing the range of motion, improving proprioception, and enhancing mental health and well-being. Additionally, tai chi is used to treat respiratory diseases and as an additional treatment for cardiovascular conditions such as hypertension and heart failure [113].

Supporting the beneficial effects of tai chi on cardiovascular function, some studies have reported decreased systolic and diastolic blood pressure in hypertensive patients and a delay in heart failure progression [10,119,121]. Moreover, other studies have suggested that tai chi reduces the serum levels of the B-type natriuretic peptide and the levels of serum lipids, including total cholesterol, total triglycerides, and low-density lipoproteins type C [119,121].

Regarding the paediatric population, the literature reports that children commonly practice tai chi as a general physical practice, aiming to decrease obesity and overweight and to improve well-being and physical fitness. Moreover, some RCTs have shown positive effects on respiratory fitness, balance, the coordination of the upper limbs, and functional mobility, particularly in patients with intellectual disabilities and children with balance disorders [118,122,123].

In conclusion, tai chi’s strongest scientific evidence in children favours its practice as a general physical activity with beneficial effects on overall fitness and body weight, particularly applicable to children with coordination disorders. Based on the results mentioned earlier regarding studies conducted in adults, the positive results in terms of impacting cardiovascular and respiratory function should be explored and evaluated in the paediatric population.

### 7.4. Herbal Medicine: Overview and Mechanism of Action

Herbal medicine is defined as all products composed exclusively of active herbal substances [124]. Medicinal treatments include various components of plants, encompassing their specific parts, such as the roots, leaves, seeds, fruits, or flowers or the dried plant [125]. Some of these components can be prepared as an infusion, tea, pill, extract, oil, or cream [125].

Herbs’ medicinal properties have been used for centuries, especially in indigenous cultures, to treat several symptoms and illnesses among children [126]. Regions such as China, India, and Egypt were pioneers in herbal medicine, and, in western countries, herbs were commonly used during mediaeval times [125].

Similar to many other therapies, the tendency to use herbal medicine is rising in some countries, such as the USA [127]. The general population’s global rate of herbal remedy use is around 80% among adults and children, with an estimated prevalence of 18% among the paediatric population [128]. According to a national USA survey in 2012, herbal remedies accounted for the most used CAM method, with 3.9% of children under CAM treatment [17].

Unlike herbal medicine, a drug is defined as any substance that enters the body and produces physiological changes that are used for prevention (e.g., vaccines, vitamins, diet) or to treat pathological conditions (e.g., hypothyroidism, malnutrition, diabetes) [129]. Taking into account this pharmacological principle, herbs cannot be prescribed as if they are an “active drug principle”, given that herbs’ components broadly differ due to several intrinsic and extrinsic factors, such as the circadian rhythm, the type of species, seasonal variations, soil and geographical conditions, and harvesting and processing methods [130].

Unlike herbal medicine, “active drug principles” have a well-defined physicochemical profile, pharmacokinetics, bioavailability, and posology. In herbal medicine, many species’ pharmacological profiles and active compounds are not properly elucidated [131]. Since herbal medicines are a collection of multiple molecules in a single plant fragment, elucidating their mechanisms of action and precise posology remains remarkably challenging [132].

Herbal medicine practitioners and academics suggest that herbs’ prophylactic and therapeutic properties are influenced by the synergistic effects of multiple herb compounds [131]. Hence, rather than using a single active drug principle, as in western medicine, the selected herb fragment should be utilised as a whole under the premise that such plant parts have greater therapeutic properties than their individual compounds [133].

#### Applications and Adverse Effects

Despite the vast number of plants and their possible properties, parents often seek herbal remedies to treat their children for conditions such as upper respiratory tract infections, cough, the “common cold”, noninfective gastroenteritis and colitis, pain in the throat and chest, asthma, cancer, neurological disorders, atopic dermatitis, abdominal pain, and prophylactic treatments for different conditions [126,134,135].

Parents usually choose herbal medicines for their children under the generic belief that they are effective and safe, because herbal remedies come from natural sources [135]. Nevertheless, many herbs’ therapeutic actions are poorly understood because some active substances have not yet been identified as chemically defined ingredients; hence, examining the properties of herbs and conducting clinical trials to test their effectiveness is complex [134]. Moreover, due to the lack of knowledge of the active substances, herbs’ dosage is based on subjective perception, signifying a potential risk of adverse effects, overdoses, and poisoning [126].

The available evidence of some commonly used herbs such as *Gingko biloba*, *Zingiber officinale* (ginger), peppermint oil, and Echinacea suggests promising and beneficial effects among the paediatric population. In some RCTs conducted on children and adolescents with ADHD using herbs such as *Gingko biloba* and *Valeriana officinalis*, it was reported that such herbs might ameliorate ADHD symptoms [136,137]. However, systematic reviews and meta-analyses assessing the possible effects of herbs on children with ADHD have found that the available evidence is inconclusive and insufficient to recommend herbs as an alternative treatment for ADHD [138,139,140,141].

Some other herbal compounds, such as peppermint oil and ginger, have been evaluated as possible treatments for gastrointestinal disorders, including functional abdominal pain, irritable bowel symptoms, and gastroenteritis, in children [141,142,143,144,145]. An RCT conducted in 75 children with acute gastroenteritis found that ginger was an effective intervention to improve vomiting compared to a placebo (*p* = 0.003) [143]. Still, in gastrointestinal disorders, an RCT assessing the effects of peppermint oil in irritable bowel syndrome reported that peppermint oil was only effective in decreasing the abdominal pain intensity compared to a placebo (*p* = 0.001) [146]. However, peppermint oil did not show significant changes when assessing belching, abdominal distention, heartburn, gas, or abdominal rumbling. Similar results were found in other RCTs, showing no significant differences in gut motility, gastric emptying, and contractility when using peppermint oil in children with functional abdominal pain [144].

Some other herbs, such as *Echinacea* spp. and *Astragalus* spp., have been used as treatment interventions for respiratory conditions in the paediatric population [147,148,149,150]. For instance, in an RCT evaluating Echinacea’s effectiveness in preventing viral respiratory tract infections in children, Echinacea appeared to be effective in reducing episodes of viral respiratory tract infection compared to vitamin C (*p* = 0.021) [148]. Another mini-review of RCTs evaluating Echinacea’s effects against coronavirus in children and adults agreed that Echinacea may exhibit antiviral activity [149].

It is worth mentioning that the modern pharmaceutical industry also relies on many herbal products and components, and many pharmaceutical products include natural plant products or analogues derived from herbs.

The significance of herbs in contemporary medicine is exemplified by former chemotherapeutic agents, such as paclitaxel [124]. This chemotherapeutic agent was first isolated from the tree *Taxus brevifolia*. Currently, compounds derived from *Taxus brevifolia* are employed in the treatment of specific types of cancer, including ovarian, breast, and pancreatic cancers, among others [124,151].

Likewise, paclitaxel—the anti-tumour alkaloid homoharringtonine, derived from an extract of *Cephalotaxus harringtonii* (Japanese plum yew)—was previously employed by Asian populations as an antimicrobial agent [124]. Currently, extracts of *Cephalotaxus harringtonii* are approved by the Food and Drug Administration (FDA) and employed in contemporary medicine to treat patients with chronic myeloid leukaemia [124]. In line with these two examples, it is estimated that, today, over 50% of synthetic and conventional medications are derived from different plant species [152,153,154].

In summary, specific herbal compounds isolated from *Taxus brevifolia* and *Cephalotaxus harringtonii* have demonstrated pharmacological efficacy. However, the results from other clinical trials reporting positive effects for herbs like *Ginkgo biloba*, *Zingiber officinale*, *Valeriana officinalis*, *Echinacea* spp., and *Astragalus* spp. should be interpreted with caution. The validity and generalisability of these results to the paediatric population are often limited by methodological heterogeneity, a lack of long-term follow-up, small sample sizes, and the absence of common clinical and objective measurements, as concluded in the available systematic reviews [138,139,140,141,155]. Further research is needed to address these knowledge gaps and establish the efficacy and safety profiles of these herbal treatments, potentially providing alternative treatment approaches for consideration in clinical practice.

### 7.5. Energy Therapy

#### 7.5.1. Reiki: Overview and Mechanism of Action

Reiki is an energy-healing or biofield therapy that originated in Japan around the late 19th century. Literally speaking, “rei” means universal and “ki” means life energy, and the term can be translated into “universal life energy” [156,157].

Reiki sustains the idea that universal energy directly supports the human body’s ability to heal; hence, this technique seeks to send energy through the therapist’s hands to the sick body, enhancing its innate ability to heal and restore its natural balance [156,158].

Given this energetical practice’s nature and characteristics, elucidating its exact mechanism of action is challenging. Studies conducted in animal models have not shown enough scientific evidence suggesting the direct induction of physiological changes by reiki practices; however, it has been proposed that reiki activates the parasympathetic nervous system through the induction of a relaxation state, resulting in reduced cortisol and a lower heart rate and blood pressure [156,159]. Still, these effects remain highly controversial as other authors have found no correlation between reductions in either heart rate or blood pressure and reiki in animal models. For instance, a study described that rats that received 15 min of reiki showed a decreased heart rate compared with rats receiving sham reiki (individuals that mimicked reiki movements without proper formation in reiki). However, no significant changes in blood pressure were found; hence, the results cannot be extrapolated to the general population [160]. Additionally, another study conducted on 189 oncologic patients found similar results for a sham reiki placebo and reiki concerning well-being and comfort, concluding that neither the sham reiki placebo nor reiki altered the physical well-being of oncological patients. However, this study has highlighted the effect of “being with”, defined as being emotionally present with others, implying that reiki therapy has a positive psychological impact on the sense and feeling of receiving extra attention, care, and privacy; as a consequence, it enhances the patient’s perception of well-being [161].

In summary, reiki’s effects on health remain controversial, and the scientific evidence yields paradoxical results; the documented psychological effects such as well-being, relaxation, and comfort outweigh the objective physiological effects of reiki.

#### 7.5.2. Reiki: Applications

In the general population, reiki therapy has been used as a treatment or as a supportive measure for several conditions, mainly within palliative care in patients with end-stage cancer and severe pain. For instance, it has been reported that reiki may have a significant effect on pain relief in the areas of dental surgery, chronic pain, abdominal hysterectomies, post-caesarean surgeries, and neuropathic pain [158].

In an RCT, it was found that patients with a diagnosis of cancer who used reiki as a supportive therapy reported a positive experience concerning mood-related symptoms, including a decrease in anxiety, reduced isolation and loneliness, a better attitude, an improved appetite, and higher-quality sleep [162,163,164]. Likewise, reiki has been used in the field of mental health and well-being, showing beneficial results in terms of decreasing depression, anxiety, and stress symptoms in non-related oncological patients [161,165].

In the paediatric population, the literature is scarce. However, some pilot studies have been conducted in children with a diagnosis of cancer and hematopoietic stem cell transplantation patients, showing promising results in reducing pain. Still, the authors concluded that there was no statistical significance in relation to objective variables; however, an overall improvement in pain perception was achieved [164,166].

In summary, reiki has not shown statistically significant effects over physiological pathways and variables, but a trend towards pain improvement and overall well-being has been suggested.

## 8. Future Perspectives and Limitations

Despite our extensive literature review, it is challenging to determine a global consensus on the integration of CAM as a single and homogeneous concept due to the internal idiosyncrasies and cultural perceptions of each country implementing CAM in their healthcare systems and the diverse public health regulations. Moreover, the lack of consensus and the polarised positions among conventional professionals regarding the mechanisms of action, applications, age groups, target populations, and effectiveness of CAM make it challenging to draw firm conclusions. This situation highlights the need for high-quality research examining the efficacy and safety of CAM therapies in vulnerable populations, such as children.

Another potential limitation of this review is the variability in the quality and methodologies of the studies included, which might complicate the direct comparison of the outcomes and generalisation of the findings. Lastly, the potential lack of long-term longitudinal studies limits our understanding of the long-term impact of CAM therapies on paediatric health, especially regarding side effects, drug interactions, and long-term health outcomes.

Therefore, obtaining more knowledge and scientific evidence of CAM’s effectiveness and safety will help to decrease the morbidity and mortality rates related to ineffective and unsafe health practices. CAM practitioners and mainstream healthcare professionals require collaborative effort with a multidisciplinary approach to improve conventional medical practices and patients’ overall well-being. This multidisciplinary and collaborative approach would allow holistic and personalised therapies to fulfil each child’s physical, psychological, and mental needs.

It has been documented that CAM therapies are rooted in ancient cultures, with a predominant influence from East Asian countries. However, nowadays, the prevalence of CAM users is increasing worldwide, particularly in western countries. This tendency has led to a shift towards a global healthcare paradigm in relation to western healthcare practices and the acceptance of CAM within formal healthcare systems.

The nature of this paradigm is characterised by controversies between different scientific perspectives regarding CAM. On the one hand, some scholars suggest that CAM “might” be helpful and safe for certain conditions if prescribed and supervised by health professionals. On the other hand, conservative scholars have suggested that CAM interventions lack scientific evidence supporting their use as an isolated or adjunctive therapy, given that the offered evidence fails to provide objective and statistically significant results over physiological or pathophysiological processes. This issue highlights the need for new research to evaluate these practices’ efficacy, safety, and cultural integration in international and national healthcare settings.

The vast majority of the existing studies assessing children and CAM have focused on particular conditions or diseases among children or within specific cultures or geographical regions. Therefore, the conclusions regarding the effectiveness and safety of CAM interventions are broad. Amongst the CAM therapies covered in this review (Table 3), acupuncture, tai chi, yoga, meditation, massage, and bodywork have stood out, given the scientific rigour and objectivity of the evidence supporting their use for the general and paediatric populations.

Evidence favouring the aforementioned therapies suggests that their positive effects are mainly related to the relaxation state and the modulation of the autonomic nervous system, which positively impact overall well-being and improve mood and behavioural conditions (e.g., anxiety, stress, insomnia, depression) and enhance pain relief. Within specific non-objective parameters, CAM therapies improve the perception of overall body shape, disease-related symptoms, comfort, mental well-being, and quality of life. However, for some of the described CAM therapies, the scientific evidence mostly fails to demonstrate tangible effects over physiological pathways and biological processes.

## 9. Conclusions

Our research highlights the need for further investigation into CAM in paediatric care, as it opens up new questions that warrant exploration. This study lays the groundwork for future inquiries, emphasising the importance of addressing critical issues such as CAM therapies’ safety, efficacy, and mechanisms of action and their integration into clinical settings. Additionally, it underscores the importance of launching scientific awareness campaigns aimed at healthcare personnel to clearly define CAM’s objective capabilities and limitations in the clinical context. It also advocates for CAM researchers and practitioners to prioritise objective and rigorous evaluation methods, thereby fostering an informed and holistic approach to paediatric healthcare. This holistic approach should integrate evidence-based practices, combining scientific evidence, the patient’s values and beliefs, and clinical expertise.

An evidence-based healthcare system, anchored in primary healthcare, directs its structures and functions towards the values of equity and social solidarity without distinction, suggesting adherence to the highest standards of quality and safety and implementing intersectoral interventions.

Furthermore, a multidisciplinary and collaborative approach must be integrated at all healthcare levels. This approach involves distinct healthcare professionals such as physicians, nutritionists, psychologists, odontologists, and CAM and traditional medicine practitioners, with active participation from paediatric patients and their guardians. Embracing diverse expertise ensures a comprehensive and inclusive healthcare strategy that addresses the holistic well-being of individuals across various medical disciplines.

In conclusion, even though there are promising results in favour of CAM, this literature review shows and concludes that the available scientific evidence is only sufficient to recommend CAM as a complementary therapy, rather than an alternative, to enhance mental well-being and physical fitness in the paediatric population.

## Figures and Tables

**Figure 1 healthcare-12-01547-f001:**
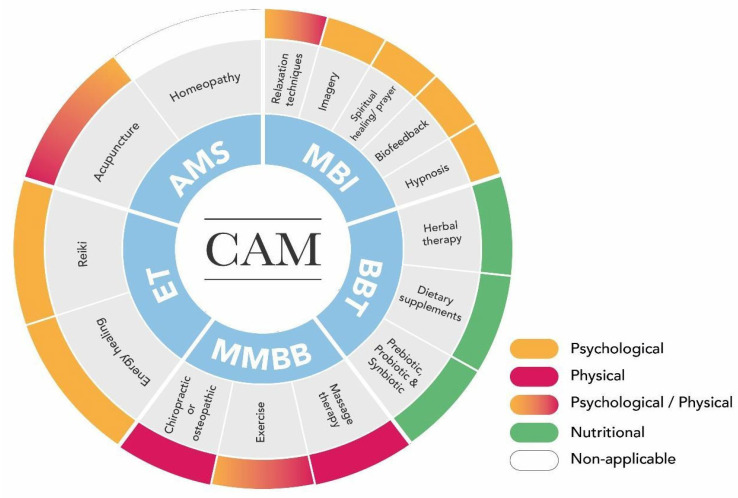
Adjusted CAM classification 2023. The above classification and categorisation were modified from Mollaoğlu and Aciyurt [61] and the NCCIH (2021) [60]. The internal circle shows the CAM class (blue), followed by the CAM subgroup (grey), and the external circle displays the CAM category. Each CAM category is further subdivided into distinct colours according to the expected effect: physiological (orange), physical (red), physiological/physical (colour gradient from orange to red), nutritional (green), and non-applicable (white). Abbreviations: AMS, alternative and medical system; MBI, mind–body interventions; BBT, biologically based therapies; MBBM, manipulative and body-based methods; ET, energy therapies.

**Table 1 healthcare-12-01547-t001:** Conceptual design of literature search.

Search Items		
**Initial terms** Complementary and alternative medicine Paediatric health Traditional medicine Healthcare system Integrative medicine	**Topical search** Acupuncture Massage and bodywork Mind–body therapies Yoga Meditation Tai chi Herbal medicine Energy therapy Reiki	**AND** Paediatric population Children Applications Mechanism of action Adverse effects
**Academic databases** Google Scholar, PubMed, and the Cochrane Library		
**Types of research** Systematic reviews, meta-analyses, randomised controlled trials (RCT), observational studies, literature reviews, case reports, and discussion articles. * Preclinical studies were included to explore mechanisms of action.		
**Language** English		
**Publication year limits** 2010–2024 ** Exceptions were made for relevant older studies or when recent data were lacking.		

**Table 2 healthcare-12-01547-t002:** Types of meditation and example descriptions.

Types of Meditation Practice	Examples
Concentration on a word, thought, sensation, or image	Transcendental meditation; relaxation response; breath-focused meditation; mantra repetition; meditation on a prayer, mandala, or other image.
Mindfulness	Mindfulness-based stress reduction, Vipassana.
Cultivating positive emotions (such as compassion, forgiveness, gratitude, or loving kindness)	Buddhist metta or tonglen practice (cultivating compassion and loving kindness), Institute of HeartMath Training (cultivating gratitude and compassion).
Emptying	Centring prayer, waiting on the inner voice or inner light.

Note: “Types of Meditation Practice” by Sibinga and Kemper, 2010. Retrieved from [108].

**Table 3 healthcare-12-01547-t003:** Summary of CAM classification and effects.

CAM Class	CAM Subgroup and Origin	Therapeutic Effect
**Alternative and medical system** **(AMS)**	Acupuncture (China)	Physical and mental
**Mind–body interventions** **(MBI)**	Yoga (India)	Physical and mental
	Tai chi (China)	Physical and mental
	Meditation (Asia)	Mental
**Biologically based therapies** **(BBT)**	Herbal medicine (Globally)	Subject to individual compounds
**Manipulative and body-based methods** **(MBBM)**	Massage therapy (China)	Physical and mental
**Energy therapies** **(ET)**	Reiki (Japan)	Mental perception

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
