# Peer review of "Complementary and Alternative Medicine (CAM) Practices: A Narrative Review Elucidating the Impact on Healthcare Systems, Mechanisms and Paediatric Applications"

_healthcare, 2024, doi:10.3390/healthcare12151547_

Round 1

Reviewer 1 Report

Comments and Suggestions for Authors

I have reviewed the manuscript entitled “Complementary and Alternative Medicine (CAM) Practices: Elucidating the Impact on Healthcare Systems, Mechanisms, and Paediatric Applications”, submitted to the Journal of Healthcare. The authors present new insights in the area of Complementary and Alternative Medicine, which is commendable. I appreciate the authors' valuable research, which can enhance the understanding of the topic and guide future research in this field. The manuscript demonstrates novelty and interest suitable for publication in the Journal of Healthcare. However, before proceeding with further evaluations, I kindly request the authors to address the comments provided below.

1.      I recommend that the authors consider engaging a professional English editor to refine and polish their manuscript. Enhancing the language and structure of the manuscript can significantly improve clarity and readability, ensuring that the research findings are effectively communicated to the readership. Attention to detail in terms of grammar, syntax, and overall coherence will further strengthen the impact of the study. I encourage the authors to prioritize this step to enhance the quality of their work and increase the manuscript's accessibility and impact.

2.      I suggest that the authors define abbreviations (e.g. OR, CI) before their first use in the manuscript. This practice enhances the readability and comprehension of the text, making it easier for readers to follow along with the content without confusion. Clearly defining abbreviations upfront will facilitate a smoother reading experience and help ensure that the key terms are understood consistently throughout the document. I recommend implementing this technique to improve the overall clarity and accessibility of the manuscript for the benefit of the readers.

3.      In the introduction section, it is essential to incorporate a significant message regarding the clinical relevance of complementary and alternative medicine. The authors should emphasize the importance of these modalities in enhancing overall health outcomes and address the growing interest and trends associated with their use in contemporary healthcare practices. I strongly recommend citing the references listed below to support and reinforce the discussion on the clinical benefits and increasing popularity of complementary and alternative medicine.
- Sharifi, Mohammad Hossein, Salman Mohammadi, Naseh Pahlavani, Ali Ghaffarian-Bahraman, Sara Darabi, Mohammad Hossein Nikoo, and Mohammad Ebrahim Zohalinezhad. "Utilization of Complementary and Alternative Medicine among Patients with Cardiovascular Disease in Iran: A Cross-Sectional Study." Traditional and Integrative Medicine (2024): 13-23.

- Chen, Shu-Ping, Su-Tso Yang, Kai-Chieh Hu, Senthil Kumaran Satyanarayanan, and Kuan-Pin Su. "Usage Patterns of Traditional Chinese Medicine for Patients with Bipolar Disorder: A Population-Based Study in Taiwan." In Healthcare, vol. 12, no. 4, p. 490. MDPI, 2024.
- Ghorat, Fereshteh, Seyed Hamdollah Mosavat, Samaneh Hadigheh, Seyed Amin Kouhpayeh, Mohammad Mehdi Naghizadeh, Ali Akbar Rashidi, and Mohammad Hashem Hashempur. "Prevalence of Complementary and Alternative Medicine Use and Its Associated Factors among Iranian Diabetic Patients: A Cross-Sectional Study." Current Therapeutic Research, Clinical and Experimental 100 (2024): 100746-100746.
- Kwon, Chan-Young. "Analysis of the Use of Korean Medicine Treatments among Children and Adolescents in South Korea: Analysis of Nationally Representative Sample." In Healthcare, vol. 12, no. 4, p. 467. MDPI, 2024.

4.      A significant limitation in interpreting various studies on the "Complementary and Alternative Medicine Use" is the inconsistency in defining these terms across different studies and the utilization of diverse tools for data collection from respondents. It is crucial to address this issue in the narrative, supported by references from the articles listed below. The discussion should delve into the challenges posed by variations in definitions and data collection methods, which can impact the comparability and generalizability of research findings in this area.
- Quandt, Sara A., Marja J. Verhoef, Thomas A. Arcury, George T. Lewith, Aslak Steinsbekk, Agnete E. Kristoffersen, Dietlind L. Wahner-Roedler, and Vinjar Fønnebø. "Development of an international questionnaire to measure use of complementary and alternative medicine (I-CAM-Q)." The Journal of Alternative and Complementary Medicine 15, no. 4 (2009): 331-339.
- Druart, Leo, and Nicolas Pinsault. "The I-CAM-FR: a French translation and cross-cultural adaptation of the I-CAM-Q." 
Medicines 5, no. 3 (2018): 72.
- Chijan, Mahsa Rostami, Mehdi Salehi, Mohadese Ostovar, Elham Haghjoo, Massih Sedigh Rahimabadi, and Mohammad Hashem Hashempur. "The I-CAM-IR: persian translation, cross-cultural adaptation and revised version of the I-CAM-Q." Traditional and Integrative Medicine (2022).
- Lee, Ju Ah, Yui Sasaki, Ichiro Arai, Ho-Yeon Go, Sunju Park, Keiko Yukawa, Yun Kung Nam et al. "An assessment of the use of complementary and alternative medicine by Korean people using an adapted version of the standardized international questionnaire (I-CAM-QK): a cross-sectional study of an internet survey." BMC complementary and alternative medicine 18 (2018): 1-11.

5.      It is recommended that in the glossary section, add Persian medicine (https://www.ncbi.nlm.nih.gov/mesh/2101249) and its description.

Comments on the Quality of English Language

Please see "Comments and Suggestions for Authors
".

Author Response

Reviewer 1

Summary: 

I have reviewed the manuscript entitled “Complementary and Alternative Medicine (CAM) Practices: Elucidating the Impact on Healthcare Systems, Mechanisms, and Paediatric Applications”, submitted to the Journal of Healthcare. The authors present new insights in the area of Complementary and Alternative Medicine, which is commendable. I appreciate the authors' valuable research, which can enhance the understanding of the topic and guide future research in this field. The manuscript demonstrates novelty and interest suitable for publication in the Journal of Healthcare. However, before proceeding with further evaluations, I kindly request the authors to address the comments provided below.

Response summary: 

We sincerely appreciate the reviewer-1’s careful evaluation and comments on improving our manuscript.   

Comment 1:  

I recommend that the authors consider engaging a professional English editor to refine and polish their manuscript. Enhancing the language and structure of the manuscript can significantly improve clarity and readability, ensuring that the research findings are effectively communicated to the readership. Attention to detail in terms of grammar, syntax, and overall coherence will further strengthen the impact of the study. I encourage the authors to prioritize this step to enhance the quality of their work and increase the manuscript's accessibility and impact.

Response 1: 

We thank the reviewer for raising up this crucial point. In response to this concern, we have carefully reviewed the grammar, syntax, and coherence of the manuscript, with particular emphasis on the abstract which was updated to improve clarity and quality. 

Comment 2:

I suggest that the authors define abbreviations (e.g. OR, CI) before their first use in the manuscript. This practice enhances the readability and comprehension of the text, making it easier for readers to follow along with the content without confusion. Clearly defining abbreviations upfront will facilitate a smoother reading experience and help ensure that the key terms are understood consistently throughout the document. I recommend implementing this technique to improve the overall clarity and accessibility of the manuscript for the benefit of the readers.

Response 2: 

Thank you for this comment. We have now defined the aforementioned abbreviations and have reviewed the rest of the included abbreviations to avoid misunderstandings. 

Comment 3: 

In the introduction section, it is essential to incorporate a significant message regarding the clinical relevance of complementary and alternative medicine. The authors should emphasize the importance of these modalities in enhancing overall health outcomes and address the growing interest and trends associated with their use in contemporary healthcare practices. I strongly recommend citing the references listed below to support and reinforce the discussion on the clinical benefits and increasing popularity of complementary and alternative medicine.

- Sharifi, Mohammad Hossein, Salman Mohammadi, Naseh Pahlavani, Ali Ghaffarian-Bahraman, Sara Darabi, Mohammad Hossein Nikoo, and Mohammad Ebrahim Zohalinezhad. "Utilization of Complementary and Alternative Medicine among Patients with Cardiovascular Disease in Iran: A Cross-Sectional Study." Traditional and Integrative Medicine (2024): 13-23.

- Chen, Shu-Ping, Su-Tso Yang, Kai-Chieh Hu, Senthil Kumaran Satyanarayanan, and Kuan-Pin Su. fyub"Usage Patterns of Traditional Chinese Medicine for Patients with Bipolar Disorder: A Population-Based Study in Taiwan." In Healthcare, vol. 12, no. 4, p. 490. MDPI, 2024.

- Ghorat, Fereshteh, Seyed Hamdollah Mosavat, Samaneh Hadigheh, Seyed Amin Kouhpayeh, Mohammad Mehdi Naghizadeh, Ali Akbar Rashidi, and Mohammad Hashem Hashempur. "Prevalence of Complementary and Alternative Medicine Use and Its Associated Factors among Iranian Diabetic Patients: A Cross-Sectional Study." Current Therapeutic Research, Clinical and Experimental 100 (2024): 100746-100746.

- Kwon, Chan-Young. "Analysis of the Use of Korean Medicine Treatments among Children and Adolescents in South Korea: Analysis of Nationally Representative Sample." In Healthcare, vol. 12, no. 4, p. 467. MDPI, 2024.

Response 3: 

Thank you for the insightful evaluation. After a careful review and based on the reviewer’s suggestion, we have added the following reference to enrich the content of section 4 (Lines ), “Paediatric Population and CAM” as well as in section 5.2 (Lines ), “Economical and educational factors.” 

  •  Kwon, Chan-Young. "Analysis of the Use of Korean Medicine Treatments among Children and Adolescents in South Korea: Analysis of Nationally Representative Sample." In Healthcare, vol. 12, no. 4, p. 467. MDPI, 2024. 

“The prevalence of CAM therapies in paediatric populations varies widely according to cultural and geographical contexts; for instance, in South Korea, where Korean medicine –a form of East Asian Traditional Medicine, plays a central role within South Korea’s healthcare system, 8.13% of the population under 19 years-old uses integrative medicine. Still, it is estimated that over the world, overall rates of CAM use in children range from 10.9% to 87.6% depending on the country and socio-cultural factors of the parents”.

Given the paediatric scope of our manuscript, we were unable to add the rest of the suggested references, which, although valuable, focus on the adult population. 

Comment 4: 

A significant limitation in interpreting various studies on the "Complementary and Alternative Medicine Use" is the inconsistency in defining these terms across different studies and the utilization of diverse tools for data collection from respondents. It is crucial to address this issue in the narrative, supported by references from the articles listed below. The discussion should delve into the challenges posed by variations in definitions and data collection methods, which can impact the comparability and generalizability of research findings in this area.

- Quandt, Sara A., Marja J. Verhoef, Thomas A. Arcury, George T. Lewith, Aslak Steinsbekk, Agnete E. Kristoffersen, Dietlind L. Wahner-Roedler, and Vinjar Fønnebø. "Development of an international questionnaire to measure use of complementary and alternative medicine (I-CAM-Q)." The Journal of Alternative and Complementary Medicine 15, no. 4 (2009): 331-339.

- Druart, Leo, and Nicolas Pinsault. "The I-CAM-FR: a French translation and cross-cultural adaptation of the I-CAM-Q." Medicines 5, no. 3 (2018): 72.

- Chijan, Mahsa Rostami, Mehdi Salehi, Mohadese Ostovar, Elham Haghjoo, Massih Sedigh Rahimabadi, and Mohammad Hashem Hashempur. "The I-CAM-IR: persian translation, cross-cultural adaptation and revised version of the I-CAM-Q." Traditional and Integrative Medicine (2022).

- Lee, Ju Ah, Yui Sasaki, Ichiro Arai, Ho-Yeon Go, Sunju Park, Keiko Yukawa, Yun Kung Nam et al. "An assessment of the use of complementary and alternative medicine by Korean people using an adapted version of the standardized international questionnaire (I-CAM-QK): a cross-sectional study of an internet survey." BMC complementary and alternative medicine 18 (2018): 1-11.

Response 4: 

We greatly appreciate all the comments made on our manuscript. After reviewing and analyzing the suggested references, we acknowledge that discussing this crucial point in our manuscript will strengthen the content and quality of the manuscript; thus, we have included some of the  references in the introduction section (Lines ): 

“Alike, differences in CAM prevalence, purpose, and satisfaction measurements, pose a significant challenge due to the variability of findings across distinct study designs and countries.”

Comment 5:     

It is recommended that in the glossary section, add Persian medicine (https://www.ncbi.nlm.nih.gov/mesh/2101249) and its description.

Response 5: 

We appreciate the reviewer's valuable insights. However, including Persian Medicine would divert the main focus of our manuscript and exceed the initially defined scope. Due to its complexity and the vast amount of evidence available, including Persian medicine requires separate, in-depth research, which deserves an independent study. This topic will surely be considered for future research.

Reviewer 2 Report

Comments and Suggestions for Authors

Submitted manuscript draws upon an increasing wealth of evidence demonstrating value and efficacy thru the use of Complementary and Alternative Medicine (CAM).  Authors astutely report that heterogenous outcomes in  youth-aged cohorts reflect markedly variable treatment philosophies for CAM use in the setting of pediatric practices.  Thus, a key facet of Author’s submitted work appears to shed light on this set of disparate data sets.  In so doing, Authors address multiple facets of CAM, including acupuncture, Yoga, Tai Chi, and herbal medicine.  Further drilling down to methodologies employed, Authors address socio-cultural, economic and educational factors.  Here, an intriguing finding reports up to 96% pediatric oncology patients attempting some form of Traditional Chinese Medicine (TCM).  

To their credit, Authors provide a quite exhaustive catalog of the various CAM methodologies in use — and they seek to touch upon evidence-based pluses and minuses.  It is here, however, where Reviewer finds a basis to raise issue with Author’s section 7. “Conclusion, Limitations & Future Perspectives”.  As articulated in lines 720 to 726, Authors state “…..CAM interventions lack scientific evidence supporting its use as an isolated or adjunctive therapy”.  Reviewer would point out that while this conclusion may well have been true, broadly speaking, throughout time and up to the 21st Century, in fact there is one key facet of CAM that has advanced dramatically — and with compelling scientific support in the form of both basic science and clinical validation.  That is, the use of botanically derived compounds, and across a spectrum of disease models.  Here, such data reflects gene marker modulation, protein expression & clinical validation.  Not only are such compounds finding utility alone, but thru thoughtful combination of materials their utility has found empirically demonstrated validation even in the setting of complex trait, polygenic phenotypes.  Moreover, such botanicals, particularly when invaginated against excretion or metabolic elimination, constitute a set of remarkably low-risk / high reward tactical therapeutic weapons.  Perhaps of even greater consequence, it is from the naturally-educed compounds that some of the most promising treatments against cancer and other such deleterious maladies are, with increasing frequency, now occurring.  

It is well to paint all CAM with a broad brush - whether discussing adult or pediatric utility.  However, when measuring naturally-based compounds against synthetic small molecules,  it is the shortsighted investigator today that entirely discounts the latter in sole favor of the former.  Reviewer believes a thoughtful reevaluation of the rapidly advancing botanical medicine field is not just warranted, but crucial to the viability of this manuscript.  

Author Response

Reviewer 2

Summary: 

Submitted manuscript draws upon an increasing wealth of evidence demonstrating value and efficacy thru the use of Complementary and Alternative Medicine (CAM).  Authors astutely report that heterogenous outcomes in  youth-aged cohorts reflect markedly variable treatment philosophies for CAM use in the setting of pediatric practices.  Thus, a key facet of Author’s submitted work appears to shed light on this set of disparate data sets.  In so doing, Authors address multiple facets of CAM, including acupuncture, Yoga, Tai Chi, and herbal medicine.  Further drilling down to methodologies employed, Authors address socio-cultural, economic and educational factors.  Here, an intriguing finding reports up to 96% pediatric oncology patients attempting some form of Traditional Chinese Medicine (TCM).  

 To their credit, Authors provide a quite exhaustive catalog of the various CAM methodologies in use — and they seek to touch upon evidence-based pluses and minuses.  It is here, however, where Reviewer finds a basis to raise issue with Author’s section 7. “Conclusion, Limitations & Future Perspectives”.  As articulated in lines 720 to 726, Authors state “…..CAM interventions lack scientific evidence supporting its use as an isolated or adjunctive therapy”.  Reviewer would point out that while this conclusion may well have been true, broadly speaking, throughout time and up to the 21st Century, in fact there is one key facet of CAM that has advanced dramatically — and with compelling scientific support in the form of both basic science and clinical validation.  That is, the use of botanically derived compounds, and across a spectrum of disease models.  Here, such data reflects gene marker modulation, protein expression & clinical validation.  Not only are such compounds finding utility alone, but thru thoughtful combination of materials their utility has found empirically demonstrated validation even in the setting of complex trait, polygenic phenotypes.  Moreover, such botanicals, particularly when invaginated against excretion or metabolic elimination, constitute a set of remarkably low-risk / high reward tactical therapeutic weapons.  Perhaps of even greater consequence, it is from the naturally-educed compounds that some of the most promising treatments against cancer and other such deleterious maladies are, with increasing frequency, now occurring.  

It is well to paint all CAM with a broad brush - whether discussing adult or pediatric utility.  However, when measuring naturally-based compounds against synthetic small molecules,  it is the shortsighted investigator today that entirely discounts the latter in sole favor of the former.  Reviewer believes a thoughtful reevaluation of the rapidly advancing botanical medicine field is not just warranted, but crucial to the viability of this manuscript.  

Response summary: 

We highly appreciate the reviewer-2’s observations. The segment in question, “…..CAM interventions lack scientific evidence supporting their use as an isolated or adjunctive therapy,” as referenced by the reviewer, represents the opinion of the conservative scholars utilized to facilitate a comparative analysis of disparate ideologies. However, it was not our specific conclusion regarding the scientific evidence supporting CAM therapies.

“The nature of this paradigm relies on controversies between different CAM scientific perspectives. On the one hand, some scholars suggest CAM "might" be helpful and safe for certain conditions if prescribed and supervised by health professionals. On the other hand, conservative scholars have suggested that CAM interventions lack scientific evidence supporting their use as an isolated or adjunctive therapy, given that the offered evidence fails to prove objective and statistically significant results over physiological or pathophysiological processes. This paradigm highlights the need for new research to evaluate these practices' efficacy, safety, and cultural integration in international and national healthcare settings.”

We acknowledge the pivotal role of herbal medicine and the rapid advance of such science. Attending this important concern, we have now included a few paragraphs exemplifying its importance in modern medicine:

Herbal medicine extension: 

“ The significance of herbs in contemporary medicine is exemplified by former chemotherapeutic agents, such as paclitaxel. The chemotherapeutic agents were first isolated from the tree Taxus brevifolia. Currently, compounds derived from Taxus brevifolia are employed in the treatment of specific types of cancer, including ovarian, breast, and pancreatic cancers, among others.  

Likewise paclitaxel, the anti-tumour alkaloid homoharringtonine, derived from the extract of Cephalotaxus harringtonii (Japanese plum yew), was previously employed by Asian populations as an antimicrobial agent. Currently, the extract of Cephalotaxus harringtonii is Food and Drug Administration (FDA)-approved and is employed in contemporary medicine to treat patients with chronic myeloid leukaemia. Similarly to these two examples, it is estimated that today over 50% of synthetic and conventional medications are derived from different plant species.  

In summary, specific herbal compounds isolated from Taxus brevifolia and Cephalotaxus harringtonii have demonstrated pharmacological efficacy. However, results from other clinical trials reporting positive effects for herbs like Ginkgo biloba, Zingiber officinale, Valeriana officinalis, Echinacea spp., and Astragalus spp. should be interpreted with caution. The validity and generalizability of these results to the pediatric population are often limited by methodological heterogeneity, lack of long-term follow-ups, small sample sizes, and the absence of common clinical and objective measurements, as concluded in available systematic reviews [130-133,143]. Further research is needed to address these knowledge gaps and establish the efficacy and safety profiles of these herbal treatments, potentially providing alternative treatment approaches for consideration in clinical practice.”

As well, modifications were made on the abstract, specifically in the herbal medicine conclusion. 

Abstract modification:

“In light of the intricate and multifaceted nature of herbal medicine, it is imperative to assess its efficacy on a case-by-case basis, taking into account the specific compounds and procedures involved.”

Reviewer 3 Report

Comments and Suggestions for Authors

Comments mentioned in the paper. 

Certain aspects of the discussion on herbal drugs necessitate modifications. There are numerous reports detailing the mechanisms of action of individual phytochemicals, their combinations, and crude extracts, particularly employing techniques such as next-generation sequencing. Therefore, stating that these compounds lack a known mechanism of action could be misleading.

Comments on the Quality of English Language

Language editing is imperative, as certain sections fail to effectively communicate the intended ideas. This refinement is especially critical in the abstract portion.

Author Response

Reviewer 3

Comment 1:

Certain aspects of the discussion on herbal drugs necessitate modifications. There are numerous reports detailing the mechanisms of action of individual phytochemicals, their combinations, and crude extracts, particularly employing techniques such as next-generation sequencing. Therefore, stating that these compounds lack a known mechanism of action could be misleading.

Response 1:
We would like to express our gratitude to Reviewer-3 for the insightful comments and efforts to improve our manuscript; according to the suggestions, the abstract was updated as follows:

“While research on Complementary and Alternative Medicine (CAM) for the general population is expanding, there remains a scarcity of studies investigating the efficacy and utilization of CAM practices specifically in the pediatric population. In accordance with the World Health Organization (WHO), the prevalence of parental utilisation of CAM in their dependents is estimated to reach up to 80%. This literature review identified broad, heterogeneous, and inconclusive evidence regarding CAM applications and effectiveness, primarily attributed to variances in sociodemographic factors and differences in national healthcare systems. Additionally, the review identified a lack of consensus and polarised positions among mainstream professionals regarding the mechanism of action, applications, and effectiveness of CAM. This narrative review presents varied results concerning the efficacy of most CAM therapies and their applications, yet some evidence suggests potential benefits for acupuncture, Yoga, Tai Chi, and massage in improving physical and mental health. Moreover, available evidence indicates that meditation may enhance mental health, while Reiki may only influence patients' perceptions of comfort. In light of the intricate and multifaceted nature of herbal medicine, it is imperative to assess its efficacy on a case-by-case basis, taking into account the specific compounds and procedures involved. This comprehensive review serves as a valuable resource for health professionals, offering guidance for personalised healthcare approaches that consider the values and beliefs of patients, thereby facilitating integrated, evidence-based practices aimed at enhancing the quality of healthcare services and patient satisfaction.”

Reviewer 4 Report

Comments and Suggestions for Authors

The authors presented a literature review evaluating the use of complementary and alternative medicine (CAM) in different countries around the world.  They reviewed what scientific evidence exists in the literature describing the effects of different CAM methods and the adverse effects of the therapies especially in children.

The layout of the chapters in the article is correct. The content of the chapters is well presented.

In the first part of the paper, the authors described the cultural differences in the perception of CAM in different countries. It is good that they wrote this at the beginning because CAM is very specific and linked to region and culture.

It is very difficult to clearly divide alternative medicine methods and to find scientific evidence for the effects of these methods. The authors did a good job of mapping the different methods and how to verify their effects.

An additional difficulty of the study was the evaluation of the effect of CAM in paediatric patients. The authors described studies from different sized study groups, at different ages. It is best to compare groups that are similar in number and age. The authors may consider this in future studies. 

They also described different study methods. This may significantly affect the description of the effectiveness of the therapy. They may need to select those articles that describe similar research methodology.

The article presented was based on a very large amount of literature. This quantity adds greatly to the quality of the assessed publication. 

Author Response

Reviewer 4

Comment 1:
The authors presented a literature review evaluating the use of complementary and alternative medicine (CAM) in different countries around the world.  They reviewed what scientific evidence exists in the literature describing the effects of different CAM methods and the adverse effects of the therapies especially in children.

The layout of the chapters in the article is correct. The content of the chapters is well presented.

In the first part of the paper, the authors described the cultural differences in the perception of CAM in different countries. It is good that they wrote this at the beginning because CAM is very specific and linked to region and culture.

It is very difficult to clearly divide alternative medicine methods and to find scientific evidence for the effects of these methods. The authors did a good job of mapping the different methods and how to verify their effects.

An additional difficulty of the study was the evaluation of the effect of CAM in paediatric patients. The authors described studies from different sized study groups, at different ages. It is best to compare groups that are similar in number and age. The authors may consider this in future studies. 

They also described different study methods. This may significantly affect the description of the effectiveness of the therapy. They may need to select those articles that describe similar research methodology.

The article presented was based on a very large amount of literature. This quantity adds greatly to the quality of the assessed publication. 

Response 1:
We would like to express our gratitude to Reviewer-4 for the constructive criticism provided. We highly appreciate the time the reviewer took to in-depth analyse our manuscript and the referenced evidence.  

We acknowledge that it would be ideal to compare groups with similar in number and age, and standardized study designs; however, due to the complexity and the vast amount of evidence available for CAM practices this concern represents a critical limitation which was highlighted in our limitations section and will surely be considered for future research.

Reviewer 5 Report

Comments and Suggestions for Authors

The review explores Complementary and Alternative Medicine (CAM) in pediatrics, highlighting a significant research gap despite WHO estimates that up to 80% of parents use CAM for their children. The evidence for CAM's effectiveness is varied and inconclusive due to sociodemographic and healthcare system differences. This review underscores the importance of personalized, evidence-based health practices. The review is well-written and informative.

The authors list the factors influencing CAM usage in the pediatric population. It would be beneficial to include topics such as psychological factors, preferences and attitudes towards CAM, the type of health issue, and symptom severity.

Comments on the Quality of English Language

English looks good

Author Response

Reviewer 5 

Comment 1:

The review explores Complementary and Alternative Medicine (CAM) in pediatrics, highlighting a significant research gap despite WHO estimates that up to 80% of parents use CAM for their children. The evidence for CAM's effectiveness is varied and inconclusive due to sociodemographic and healthcare system differences. This review underscores the importance of personalized, evidence-based health practices. The review is well-written and informative.

The authors list the factors influencing CAM usage in the pediatric population. It would be beneficial to include topics such as psychological factors, preferences and attitudes towards CAM, the type of health issue, and symptom severity.

Response 1: 

We highly appreciate the reviewer-5’s insightful evaluation to our manuscript. 

Round 2

Reviewer 2 Report

Comments and Suggestions for Authors

Authors' follow up to Reviewer's critique constitutes a good faith effort to address stated concerns.